# Photodynamic Activity of Acridine Orange in Keratinocytes under Blue Light Irradiation

Bárbara Fornaciari [1], Marina S. Juvenal [1], Waleska K. Martins [2], Helena C. Junqueira [1] and Maurício S. Baptista [1,*]

1   Department of Biochemistry, Institute of Chemistry, University of São Paulo, São Paulo 05508-000, SP, Brazil
2   Department of Biotechnology, Anhanguera University of São Paulo, São Paulo 05145-200, SP, Brazil
*   Correspondence: baptista@iq.usp.br

**Abstract:** Acridine orange (AO) is a metachromatic fluorescent dye that stains various cellular compartments, specifically accumulating in acidic vacuoles (AVOs). AO is frequently used for cell and tissue staining (in vivo and in vitro), mainly because it marks different cellular compartments with different colors. However, AO also forms triplet excited states and its role as a photosensitizer is not yet completely understood. Human immortalized keratinocytes (HaCaT) were incubated for either 10 or 60 min with various concentrations (nanomolar range) of AO that were significantly lower than those typically used in staining protocols (micromolar). After incubation, the cells were irradiated with a 490 nm LED. As expected, cell viability (measured by MTT, NRU and crystal violet staining) decreased with the increase in AO concentration. Interestingly, at the same AO concentration, altering the incubation time with HaCaT substantially decreased the 50% lethal dose ($LD_{50}$) from 300 to 150 nM. The photoinduced cell death correlated primarily with lysosomal disfunction, and the correlation was stronger for the 60 min AO incubation results. Furthermore, the longer incubation time favored monomers of AO and a distribution of the dye to intracellular sites other than lysosomes. Studies with mimetic systems indicated that monomers, which have higher yields of fluorescence emission and singlet oxygen generation, are favored in acidic environments, consistent with the more intense emission from cells submitted to the longer AO incubation period. Our results indicate that AO is an efficient PDT photosensitizer, with a photodynamic efficiency that is enhanced in acidic environments when multiple intracellular locations are targeted. Consequently, when using AO as a probe for live cell tracking and tissue staining, care must be taken to avoid excessive exposure to light to avoid undesirable photosensitized oxidation reactions in the tissue or cell under investigation.

**Keywords:** photodynamic therapy; visible light; keratinocytes; subcellular localization; singlet oxygen; dimers

## 1. Introduction

Photodynamic therapy (PDT) is a clinical modality that provokes light-activated cell/tissue death based on the incorporation of photochemically active molecules (photosensitizer, PS) into the diseased tissue followed by activation via photon absorption in the presence of oxygen [1] The PS excited states can undergo both Type-I and Type-II photosensitized oxidation reactions [2]. Type-I is a contact-dependent mechanism involving charge-transfer interaction of the excited state of the PS with biomolecules, triggering radical-chain reactions that can further produce reactive oxygen species (ROS) such as the superoxide anion ($O_2\bullet-$), hydrogen peroxide ($H_2O_2$) and hydroxyl radical ($\bullet OH$). Type-II processes involve energy transfer from the triplet excited state of the PS to molecular oxygen, resulting in the formation of singlet oxygen ($^1O_2$) and the deactivation of the PS to its ground-state [3].

Both Type-I and Type-II mechanisms cause oxidation of biological targets, but the efficiency of cell death depends on myriad of other factors [3]. The subcellular localization

of the PS is especially important because specific damages in key cellular compartments can activate several regulated cell death mechanisms [3–5]. Depending on its physical and chemical characteristics, the PS can accumulate in different organelles inside the cell, such as mitochondria [6], lysosomes [7], the plasma membrane [8], nucleus [9], or endoplasmic reticulum [10]. The photoinduced damage can trigger specific signaling pathways depending on the intracellular location(s) of the PS [11]. Equilibria of the PS in solution or in the cytosol, such as acid-base equilibria, non-covalent binding to biological targets and PS dimerization or aggregation, can also substantially affect its photodynamic efficiency [3].

AO is a metachromatic fluorescent dye that easily penetrates cell membranes in its neutral form and permeates and stains many intracellular locations. In its protonated form, it tends to accumulate in acidic organelles; therefore, AO is widely used as a marker of acidic vacuoles (AVOs), i.e., lysosomes and endosomes, which can be generically referred to as the endo-lysosomal (EL) system. Because AO has acid-base and aggregation equilibria that affect its absorption/emission properties, it stains different organelles with different colors [12–15] (Figure 1).

AO was used in one of the first reports of photodynamic action against the parasite paramecium [16]. Indeed, many studies have reported the efficacy of AO as a PDT PS [15–18]. However, its photo-cytotoxicity mechanisms are still not well understood. In aqueous solution, AO participates in acid-base ($pK_a$ = 10.25) and aggregation (dimerization with $K_D$ = 14,000 M$^{-1}$, but with trimers and tetramers also being present) equilibria [19–22], which can severely affect its photodynamic efficiency [20–25]. Moreover, because AO can stain most all intracellular locations, the primary loci of cell damage by AO are not yet known. In the present work, we examine the manner in which differences in the aggregation status and in the sites of intracellular localization affect the efficiency of photoinduced cell death caused by AO. To avoid the consequences of varying the AO concentration, since AO can bind to both high and low-affinity binding sites, we also evaluated the dependence of the photodynamic action on the incubation time in a cell culture of HaCaT keratinocytes. The use of this cell line was preferred since AO can damage lysosomal homeostasis, and hence might hamper autophagy. In this case, it would be primordial to evaluate this phenomenon with and without AO treatment in a cell line that has high levels of basal autophagic flux under normal conditions.

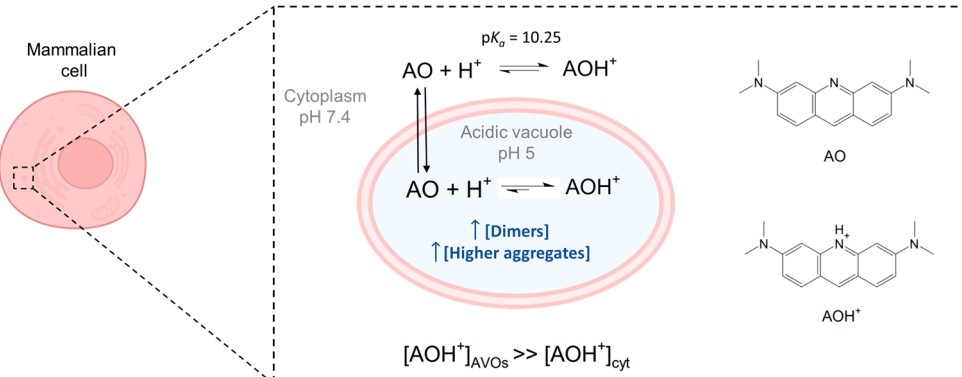

**Figure 1.** Acridine orange is a basic dye, with a $pK_a$ of 10.25 [23]. Although predominantly present in the protonated form (AOH$^+$) inside mammalian cells, its neutral form (AO) is responsible for the passage of AO through biological membranes. Due to the lower pH inside acidic vacuoles (AVOs), accumulation of the protonated structure is favored, with the result that its local concentration in AVOs is higher than that in the cytoplasm (cyt). A higher local concentration may also favor the formation of dimers and higher aggregates in AVOs. Image created with Biorender.com (accessed on 9 December 2022).

## 2. Materials and Methods

### 2.1. Chemicals

Acridine orange [3,6-bis(dimethylamino)acridinium chloride hemi(zinc chloride salt)] was obtained from Across Organics (Geel, Belgium). 3-(4,5-Dimethylthiazol-2-yl)-2,5-diphenyl tetrazolium bromide (MTT), Triton X-100, neutral red (NR), chloroquine diphosphate salt (CQ), and sodium dodecyl sulphate (SDS) were purchased from Sigma-Aldrich (St. Louis, MO, USA). Acetic acid, ethanol, sodium acetate, sodium citrate, dimethyl sulfoxide (DMSO), sodium dihydrogen phosphate, sodium hydrogen phosphate, methylene blue (MB), crystal violet (CV), and sodium citrate were purchased from Synth (Diadema, SP, Brazil). Water was deionized and purified with a Milli-Q system (Merck). AO aqueous solutions were prepared on the same day of analysis, using a molar extinction coefficient of 31,000 $M^{-1}$ $cm^{-1}$ at 489 nm for calculations.

### 2.2. Cell Culture

The human nonmalignant immortalized keratinocyte cell line (HaCaT), which was generously donated by Prof. Hugo Aguirre Armelin (Butantan Institute, Sao Paulo, Brazil), at passage 3 [25], was cultivated in Dulbecco Modified Eagle's Medium (DMEM), supplemented with 10% (*v/v*) fetal bovine serum (FBS) and 1% (*v/v*) penicillin/streptomycin. Cells were cultivated at 37 °C under moist atmosphere and 5% of $CO_2$. The viability of keratinocytes was characterized by three different assays: MTT reduction, neutral red uptake (NRU) and crystal violet staining, to access different cell compartments, i.e., mitochondria, AVOs, and cell density, respectively. HaCaT cells were chosen as a model for monitoring the interruption of or damage to autophagy, since one of the primary goals of this study was to identify whether AO was interrupting the regular autophagic flux of cells and thereby promoting cell death through an autophagy-associated mechanism. For this purpose, a non-malignant cell line with high basal levels of autophagic flux was selected; nonetheless, data from previous studies by our group support the idea that the findings reported in this manuscript can be applied to other cell lines [24,25].

### 2.3. Incorporation Assay

Exponentially growing HaCaT cells were seeded with 200 µL of a $2 \times 10^5$ cells/mL density suspension in black 96-well microplates with a clear bottom, prepared in DMEM 10% FBS (*v/v*). After 24 h of incubation at 37 °C with 5% $CO_2$, cells were treated with 100 µL of AO solutions prepared in phosphate saline buffer (PBS) with final concentrations of 200, 300 or 400 nM, and incubated for 10 or 60 min. After incubation, 50 µL of the supernatant of each well were transferred to a new plate and 50 µL of Triton 10% (*v/v*) solution added to all wells in the new plate. Wells containing cells were washed with PBS and 100 µL of Triton 5% (*v/v*) then added to the plate. Controls were treated only with PBS and were subjected to the same steps. The fluorescence of lysed cells and of the corresponding supernatant were measured with excitation at λ = 490 nm and emission at λ = 530 nm. The percentage of incorporation of AO in HaCaT cells was calculated through Equation (1).

$$\text{Incorporation of AO (\%)} = \frac{F_{lysed\ cells}}{F_{lysed\ cells} + \left(2 \times F_{supernatant}\right)} \tag{1}$$

where $F_{lysed\ cells}$ is the fluorescence intensity of cells lysed and $F_{supernatant}$ is the fluorescence intensity of AO that was not incorporated. The value of $F_{supernatant}$ was corrected, because the volume of the supernatant is diluted to the same final volume (thus the multiplication by a factor of 2). The final concentration of Triton was 5% (*v/v*) in both cases and the values of $F_{lysed\ cells}$ were corrected according to the total protein quantification in the sample using a bicinchoninic acid (BCA) protein assay (Pierce BCA Protein Assay Kit, ThermoScientific).

### 2.4. Irradiation Protocol and Cell Viability

Exponentially growing HaCaT cells were seeded with 200 μL of a $1 \times 10^5$ cells/mL suspension in 96-well clear microplates, prepared in DMEM 10% FBS (*v/v*), for 24 h. Cells were treated with AO solutions prepared in PBS at final concentrations of 200, 300 or 400 nM (controls contained PBS only) and incubated for 10 or 60 min. After incubation, the cells were washed and irradiated in PBS using a light-emitting diode (LED) irradiator ($\lambda_{exc}$ = 490 nm) (BioLambda). The light dose was 2 J/cm$^2$ and the irradiance was 1.96 mW/cm$^2$, at room temperature. After 48 h, the cells were washed and 200 μL of MTT, crystal violet or NR solutions were added, respectively, and each assay was carried out independently, similar to previous reports [25]. Briefly, MTT (50 μg/mL) and NR (30 μg/mL) solutions were prepared in DMEM 1% FBS (*v/v*) and deionized water, respectively, added to the plate and incubated for 2 h at 37 °C, 5% CO$_2$. Following this step, the solutions were removed and the cells were washed with PBS before MTT was eluted with DMSO, and absorbance values recorded at 550 nm; NR was eluted with ethanol-acetic acid solution and absorbance recorded at 540 nm. After elution, the cells were washed twice with distilled water and the same plates were stained with crystal violet (0.02%) for 5 min, at room temperature. Then, crystal violet was eluted with ethanolic sodium citrate solution and absorbance was quantified at 585 nm. All absorbances measurements were recorded with a SpectraMax i3x microplate reader (Molecular Devices). It is important to mention that the light doses used in this work are comparable to those used in PDT treatments. Thus, for example, methylene blue was successfully used at 4.5 J/cm$^2$ for cell-killing human breast cancer cells [26], rose Bengal was irradiated with 1.6 J/cm$^2$ for activation of autophagy in cervical adenocarcinoma cells [27], and Photofrin® and Photogem® in melanoma cells irradiated with blue, green and red light in the range of 0.3 to 2.4 J/cm$^2$ exhibited cytotoxic PDT effects [28]. It should be noted that blue-light is the irradiation source most used for AO-PDT protocols [29].

### 2.5. Chloroquine Treatment

After irradiation was terminated, the cells were divided into two groups: the CQ(+) group consisted of cells that received an additional CQ treatment with a final concentration of 20 μM CQ in 1% DMEM, for 24 h. The cells in the second group, CQ(−), were incubated for the same period with 1% DMEM without CQ. After incubation, the cells were washed and incubated in DMEM 10% FBS (*v/v*) for 24 h. This procedure was conducted for all treatment doses (controls—no AO, and 200, 300 or 400 nM of AO for the treated cells).

### 2.6. Singlet Oxygen Detection

Singlet oxygen ($^1\Delta_g$, $^1O_2$) phosphorescence was detected at 1270 nm using a NIR-PMT system (Hamamatsu) and excited with a LED module at 450 nm. The intensity of $^1O_2$ emission was calculated by integrating the signal of the transient decay of luminescence from the samples. For quantifying the singlet oxygen quantum yield ($\varphi_\Delta$) of AO, the area under the emission spectra was integrated relative to MB as reference compound, using Equation (2). Both dyes were dissolved in ethanol and set to have equivalent absorbances (0.10) at 450 nm. A is the absorbance at 450 nm; I is the intensity of the $^1O_2$ phosphorescence signal detected at 1270 nm. AO index stands for acridine orange and MB for methylene blue.

$$\varphi_{\Delta,AO} = \frac{A_{AO}}{A_{MB}} \times \frac{I_{AO}}{I_{MB}} \times \varphi_{\Delta,MB} \tag{2}$$

### 2.7. Spectroscopic Analyses

Stationary absorbances of AO isotropic solutions were recorded on a Shimadzu UV 1800 Spectrophotometer (400–800 nm) and fluorescence with a Varian Cary Eclipse Fluorimeter (470–700 nm), respectively, using 1 cm path length quartz cuvettes. Fluorescence emission spectra were determined by exciting the solutions at 450 nm. AO final solutions

were prepared based on the absorption spectra of stock solutions of ~10 μM. All stock solutions, as well as the final solutions, were always freshly prepared and used within 24 h.

### 2.8. Cell Fluorescence Assays

Exponentially growing HaCaT cells were seeded with 1 mL of a $1.5 \times 10^4$ cells/cm$^2$ density suspension in 24-well microplates in DMEM 10% FBS (*v/v*) and incubated for 52 h for attachment. Then, DMEM medium was removed, and the cells were washed twice with PBS and then 200 μL of DMEM was added containing LysoTracker Deep Red (LDR) at a final concentration of 100 nM. After 30 min, the medium was removed, the cells were washed twice with PBS, and incubated with 200 μL of DMEM with AO at 400 nM final concentration, for 10 min. Next, the medium was removed and the cells were washed twice with PBS. Immediately after the end of the treatment, the cells were subjected to data acquisition. In some experiments, the revelation of the effects of AO photosensitization were also done by using AO in classical cell staining protocols (1–3 μM).

Fluorescence microscopy images were obtained with a Zeiss Axiovert 200 epifluorescence inverted microscope, equipped with C-APOCHROMAT 40X/1.20 W Corr M27 (Zeiss) objective, using filters FS02 (Excitation G365, Emission LP420), FS09 (Excitation BP450-490, Emission LP515) e FS20 (Excitation BP546/12, Emission BP575-640) (Carl Zeiss, Oberkochen, Germany).

### 2.9. FLIM Images

For subcellular localization and fluorescence lifetime imaging microscopy (FLIM), 1 mL of a $1.5 \times 10^4$ cells/cm$^2$ suspension, prepared in DMEM 10% (*v/v*) FBS and 1% (*v/v*) antibiotics, was plated in a 24-well microplate. After 52 h, the DMEM medium was removed and the cells were washed twice in PBS, and then incubated for 10 or 60 min in 200 μL of DMEM medium containing AO at a final concentration of 400 nM. Fluorescence microscopic images were obtained on a Zeiss Axiovert 200 inverted epifluorescence microscope equipped with a C-APOCHROMAT 40X/1.20 W Corr M27 objective (Zeiss), using filter sets FS02 (Excitation G365, Emission LP420), FS09 (Excitation BP450-490, Emission LP515) and FS20 (Excitation BP546/12, Emission BP575-640). A MicroTime 200 time-resolved fluorescent confocal microscope (PicoQuant, Berlin, Germany) was used to obtain FLIM images. A ps pulsed diode laser (LDH-P-C-405, PicoQuant, Berlin, Germany) was used as the excitation source, operating at 405 nm, with a pulse width of >64 ps. Time-resolved luminescence was collected in Time-Correlated Single-Photon Counting (TCSPC) mode, using a TimeHarp 260 PICO plate (PicoQuant, Berlin, Germany). In all experiments, laser power was adjusted to achieve an average photon count rate of $\leq 10^5$ photons/s and maximum rates close to $10^6$ photons/s in imaging, thus significantly below the maximum count rates allowed by electronics of the TCSPC to avoid cumulative effects. Data analysis and acquisition were performed using the SymPhoTime64 Software (PicoQuant, Berlin, Germany). In this way, all photons collected in the full-frame image were used to form a global histogram for adjusting the luminescence decay. For the deconvolution fits, the instrument response function (IRF) was measured daily, recording the backscatter of the excitation light. A high-quality fit of the luminescence decay was estimated through randomly distributed residuals and low reduced chi squared values.

### 2.10. Viability Test with Sodium Azide (NaN$_3$)

Exponentially growing HaCaT cells were cultivated in Dulbecco Modified Eagle's Medium (DMEM) supplemented with 10% (*v/v*) fetal bovine serum (FBS) and 1% (*v/v*) penicillin/streptomycin, at 37 °C under moist atmosphere and 5% of CO$_2$. For the viability test, keratinocytes were seeded with 300 μL of a cell suspension of $1 \times 10^5$ cells/mL density in 48-well microplates, and incubated for 24 h for adhesion. Then, cells that received treatments were divided into 2 groups: (1) cells treated only with AO and (2) cells treated with AO and NaN$_3$. For the former group, cells were incubated with AO solution prepared in PBS, at final concentrations of 400 nM, for 60 min, before irradiation. For the latter group,

solutions of $NaN_3$ were prepared in PBS at 5, 10, and 50 mM (final concentrations), added to the cells and incubated for 15 min, prior to the irradiation protocol. Then, solutions were removed from the wells and a subsequent incubation with AO and $NaN_3$ was performed for 60 min in PBS (400 nM as final dye concentration, and 5, 10, and 50 mM as final $NaN_3$ concentrations). After incubation, the solutions were removed, cells were washed with PBS, and 300 μL of fresh PBS was added to each well for blue-light irradiation. During the irradiation step, cells in group (1) were kept only in PBS and group (2) were kept in $NaN_3$ solutions of PBS at the mentioned final concentrations (5, 10, and 50 mM), without AO. Blue light irradiation was performed with a LED box irradiator (BioLambda), at 490 nm, with a light dose of 2.00 J/cm$^2$ and irradiance of 1.96 mW/cm$^2$. Dark plates were treated with the same protocol, but without light irradiation. After photosensitization, PBS and $NaN_3$ solutions were removed from the wells, cells were washed with PBS and all wells were incubated in 10% DMEM (*v/v*) FBS for 24 h. Then, MTT solution was prepared at a final concentration of 0.5 mg/mL in 1% DMEM (*v/v*) FBS, diluted from a stock solution of 5 mg/mL (prepared in deionized water). The volume of MTT solution added to each well was 300 μL, and cells were further incubated for 2 h. After formazan crystals were formed, DMEM media was removed, cells were washed with PBS, and then 300 μL of dimethyl sulfoxide (DMSO) was added to each well for elution. The absorbances were recorded at 550 nm using a SpectraMax microplate reader, and cell viability was calculated through the ratio of absorbances of treated cells compared with untreated cells (control).

### *2.11. Statistical Analyses*

Statistical analyses were performed using IBM SPSS Statistic Software. The statistical variances between groups of samples were performed via one-way analysis of variance (ANOVA). Data are presented as mean value ± standard deviation (SD) from three independent experiments. Statistically significant values were considered those with *p*-values lower than 0.05 (*), lower than 0.01 (**) and lower than 0.001 (***).

## 3. Results and Discussion

AO (Table 1) has strong electronic absorption in the visible range, with a long-wavelength peak centered at 489 nm, corresponding to the absorption of its monomeric protonated form (AOH$^+$, p$K_a$ = 10.25) [30]. In the millimolar and sub-millimolar regimes, AO is in equilibrium with different aggregated species, whose absorption spectra are blue shifted compared with the monomeric species. For example, the characteristic absorption peak of the dimers in an AO solution is at 470 nm, while that of the monomers is at 490 nm. The maximum of the fluorescence emission spectra is always at 530 nm (independent of whether monomers or dimers are excited), because the emitting species is always the monomer [31]. AO also has a high fluorescence quantum yield ($\varphi_F$), (0.46, Table 1), with an emission maximum in the green ($\lambda_{em}$ = 530 nm). The efficiency of fluorescence emission is dependent on both acid-base and aggregation equilibria, being higher for AOH$^+$ [32,33]. The singlet excited state, $^1$PS*, can be converted to the triplet state, $^3$PS*, with a reported triplet quantum yield ($\varphi_T$) of around 0.1 (Table 1). By comparing the $^1O_2$ emission at 1270 nm of an AO solution with a solution of a known standard (MB, singlet oxygen quantum yield, $\varphi_\Delta$ = 0.52 in air-equilibrated ethanol [34]), we estimated the value of $\varphi_\Delta$, for AO to be 0.15 ± 0.01 (Figure S1). This yield is very similar to that reported by Cáceres and collaborators ($\varphi_\Delta$ = 0.18 ± 0.01) [35] and higher than expected, considering the $\varphi_T$ = 0.1, suggesting that the reported value of $\varphi_T$ may be underestimated.

**Table 1.** Photophysical parameters of AO measured at 298 K.

| $\lambda_{abs}$ (nm) | $\varepsilon_{489}$ (M$^{-1}$ cm$^{-1}$) | $\lambda_{em}$ (nm) | $\varphi_F$ | $\varphi_{IC}$ | $\varphi_T$ | $\varphi_\Delta$ |
|---|---|---|---|---|---|---|
| 489 | 31,000 [a] | 530 | 0.46 [b] | 0.44 [b] | 0.10 [b] | 0.15 |

[a] From Ref. [36]. [b] From Ref. [32].

The photodynamic activity of AO had been tested both as free dye and in liposomal suspensions, showing robust photocytotoxicity against a variety of cell types [16–18,37]. It is not trivial to determine whether AO is favoring Type I or Type II photosensitized oxidations inside cells. However, we can qualitatively compare the roles of these two mechanisms by using quenchers of singlet oxygen ($^1O_2$), such as sodium azide (NaN$_3$) [38]. Therefore, the viability of HaCaT cells incubated for 60 min with 400 nM of AO was evaluated in the presence of increasing concentrations of NaN$_3$ (Supplementary Materials-Figure S2). As expected, NaN$_3$ is toxic to the cells, significantly decreasing the cell viability in the dark. Nevertheless, the level of dark toxicity at 50 mM of NaN$_3$ (~55%) still allowed the investigation of the photoinduced toxicity. Interestingly, the viability of photosensitized cells substantially increased with the increase in NaN$_3$ concentration (viability varies from 10% at zero NaN$_3$ to 55% at 50 mM NaN$_3$), at constant AO concentration and incubation time. Note that at 50 mM NaN$_3$, the light-induced and the dark toxicities are basically the same, indicating that under this condition AO is no longer acting as a photosensitizer and quenching of $^1O_2$ completely protected HaCaT cells from AO-photoinduced damage. Therefore, Type-II photoinduced processes are indeed a major contributor to the cell death by AO (Figure S2). This is in accord with results obtained by other groups. For example, by quantifying photodamage in tRNA by AO, Amagassa (1986) showed enhancement and inhibition of tRNA photodamage in the presence of D$_2$O and NaN$_3$, respectively [39]. Zdolzek (1993) applied AO-PDT to J-774 cells and showed that pre-treatment with NaN$_3$ increased cell viability [40]. Therefore, there appears to be a strong correlation between AO photodamage and the Type-II mechanism of photosensitized oxidation.

The AO concentrations employed for staining cells are usually in the micromolar range (1–3 µM). In this work, we have used ~10-fold lower concentrations (200–400 nM) to characterize the photodynamic efficiency of AO [5,25,41]. In this concentration range, AO does not affect cell viability in the dark (Figure S3). However, after irradiation with a 490 nm LED, cell viability decreases significantly, as quantified by the decrease in lysosomal and mitochondrial function (Figure 2A), as well as by the decrease in the incorporation of crystal violet (Figure 2B). The longer AO incubation time (60 min versus 10 min) caused a significant increase in the AO photo-cytotoxicity (Figure 2). Note that at 400 nM AO there was a large difference in cell survival when comparing the 60 min with the 10 min incubation time ($p < 0.001$). After 60 min incubation, mitochondrial and lysosomal impairments were ~83% and ~85%, respectively, compared with 20 and 28%, respectively ($p < 0.001$), after a 10 min incubation (Figure 2C). There is a stronger correlation between cell viability and AO concentration at 60 min incubation. For example, to attain approximately 60% cell survival ($p = 1.0$), it is necessary to incubate HaCaT cells with 400 nM AO for 10 min or 200 nM AO for 60 min (Figure 2D). The LD$_{50}$ observed after 60 min incubation was half (~150 nM) that observed after 10 min incubation (~300 nM) (Figure S4).

Changes in cell morphology after photodamage (200 and 400 nM AO and incubation times of 10 and 60 min) were examined by staining cells with a classical AO protocol (2.5 µM) (Figures S5 and S6). Irradiation of cells after a 60 min incubation period caused severe damage to HaCaT cells, inducing a phenotype characteristic of uncontrolled cell death, while the irradiation of cells previously incubated for 10 min with AO, caused a visually less severe damage (Figures S5 and S6). It is possible to observe significant differences in the nuclei of cells that underwent photooxidation after 10 and 60 min incubations. In cells incubated for 10 min with AO at 200 nM (Figure S5E) and 400 nM (Figure S5F) and then irradiated, there is significant chromatin condensation, which indicates that cells are dying by apoptosis. In the images of cells incubated for 60 min with 200 nM of AO and then irradiated (Figure S6E), some cells showed chromatin condensation, but the vast majority exhibited disintegrated chromatin, lacking cell structure. When cells were incubated for 60 min with 400 nM of AO and irradiated (Figure S6F), the diffuse staining pattern indicate cell death, possibly by unregulated necrosis.

Although AO is a vital dye that stains the whole cell, it is a well-known lysosomotropic agent, since it protonates and accumulates in the acid environment of lysosomes [42,43].

Cells marked with AO or with LDR and their respective colocalization profile are shown in Figure S7. Although the whole cell becomes stained by AO (Figure S7A), AO accumulates in intracellular puncta that co-localize with those stained by LDR (Figure S7B,C). We (see data above) and others [16–18,37,39,40] have provided irrefutable evidence that AO also acts as a photosensitizing agent; therefore, it is natural to assume that it may cause lysosomal impairment and inhibition of the autophagic flux [41]. Although AO has been shown to efficiently stain and reveal activation or inhibition of autophagy [42], it was never shown to induce blockage of the autophagy flux. Note that 48 h after photosensitization with AO, there is a large accumulation of acid vacuoles (AVOs, Figure 2E), after either a 10 min incubation with 400 nM AO or a 60 min incubation with 200 nM AO, corroborating the results obtained by the NRU colorimetric assay (Figure 2C,D) that indicate blockage of the autophagic flux.

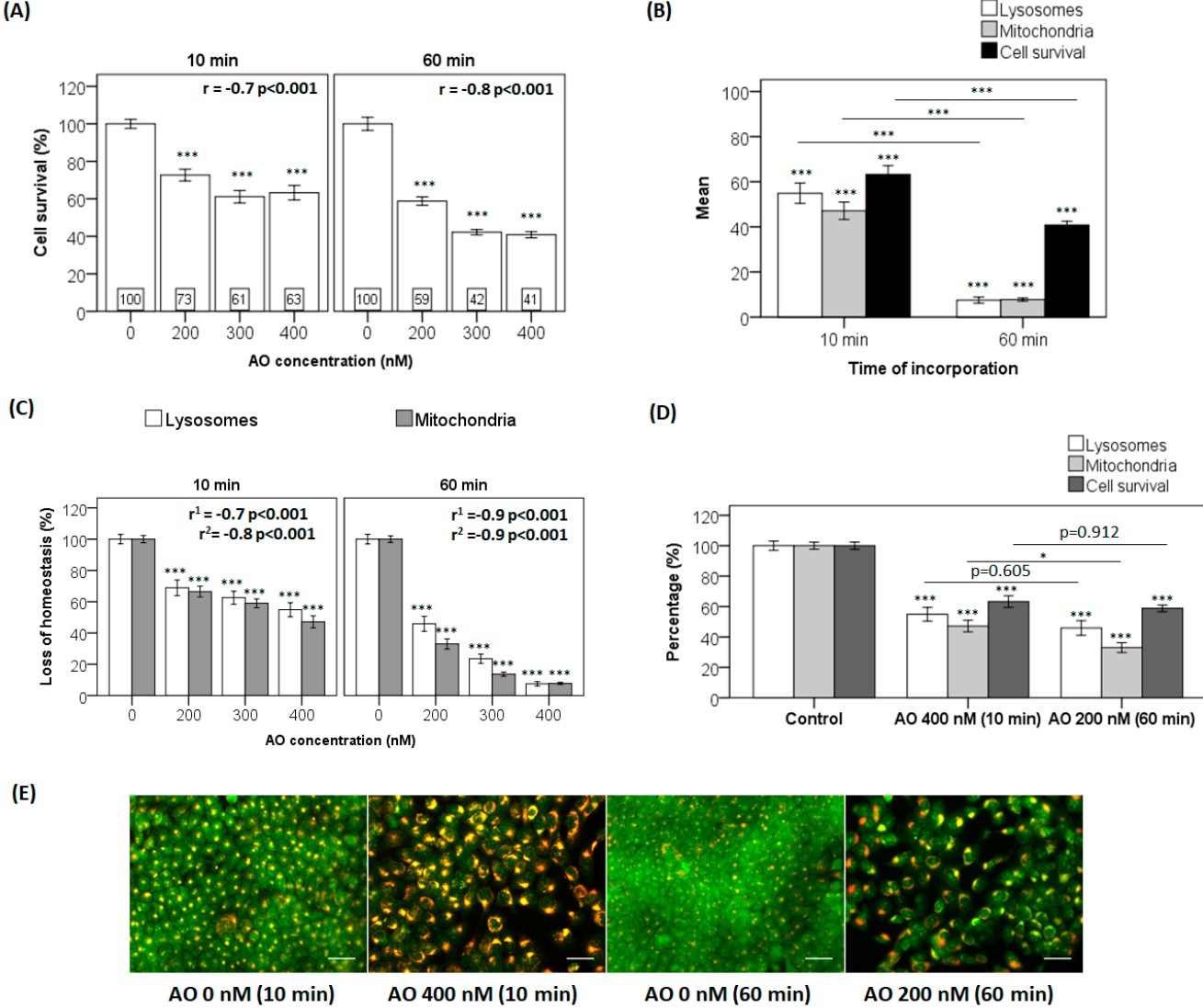

**Figure 2.** (**A**) Cell survival assay according to crystal violet. (**B**) Lysosomal homeostasis according to the NRU assay, mitochondrial homeostasis according to the MTT assay, and cell survival according to the crystal violet assay after treatment with 400 nM AO with 10 or 60 min incubation times. (**C**) Cell homeostasis by NRU and MTT for 10 or 60 min of incubation, with increasing AO concentration (0–400 nM). (**D**) Evaluation of AO treatment protocols at 10 min (400 nM) and 60 min (200 nM) for comparison of cell response in terms of lysosomal and mitochondrial homeostasis and cell survival by crystal violet. (**E**) Epifluorescence microscopy for cells stained with AO for acidic vacuoles 48 h after photodamage. Scale bar is 50 μm. ANOVA analyzes: * $p < 0.05$, *** $p < 0.001$.

To better correlate lysosome damage with the status of the autophagy flux, we chose to use HaCaT cells, since their basal autophagic levels are always active [44]. Modulation of the autophagy flux is a novel approach for targeting cancer cells [41,45]. Thus, understanding AO's impact on the autophagy levels is of paramount importance. Furthermore, our group has already applied PDT assays to both tumorous cell lines (HeLa) and keratinocytes (HaCaT) and the overall responses to our PDT protocols showed similar results [24,25].

The occurrence of parallel damage in lysosomes and mitochondria may cause autophagy blockage as an efficient cell death pathway [41]. To evaluate this premise, photosensitized cells were subjected to lysosomal inhibition by chloroquine (CQ) in parallel to the photosensitization with AO (Figure 3). Note that CQ itself induces a small decrease in cell viability (Figure 3A,B, AO(−)/CQ(+) bar). More importantly, CQ did not affect AO photosensitization at either incubation time, i.e., the survival of cells photosensitized in the presence (CQ+) and absence (CQ−) of chloroquine was the same, indicating that the lysosomes had already been photodamaged by AO before the CQ treatment ($p = 1$) (Figure 3A) [22,41]. When evaluating lysosomal homeostasis, it was observed that CQ significantly decreases it, by 30% compared to the control ($p < 0.001$). However, the lysosomal damage induced by AO photosensitization did not change in the presence or absence of chloroquine (Figure 3B). In addition, lysosomal impairment correlated with the decrease in cell survival for both incubation times (10 min, r = 0.7, $p < 0.001$; and 60 min, r = 0.8, $p < 0.001$) (Figure 3C). As expected, the lysosomal photodamage resulted in accumulation of AVOs, possibly due to autophagic inhibition (Figure 3D). Although both incubation times led to photodamage in the lysosomes of HaCaT cells, the severity of the photodamage was much higher after the 60 min incubation (Figures 2, 3, S3 and S6). Different experimental tools were thus employed in order to understand the different photocytotoxicities of AO observed after the incubation times of 10 and 60 min.

*Explaining the Photocytotoxicity of AO at Different Incubation Times*

While AO is known to photosensitize cells under light exposure [16–18], little mechanistic information is available. Understanding the mechanistic reasons for the differences in the photocytotoxicity of AO at the same initial extracellular concentration, but at different incubation times, is an interesting opportunity to understand the factors, other than the initial photosensitizer concentration, that affect the photodynamic activity of AO. The first aspect that should be considered is the amount of AO that is actually incorporated into the intracellular environment. If the 60 min incubation time allowed a higher incorporation of AO, this could obviously explain the greater photocytotoxicity of AO at the longer incubation time. For this purpose, we evaluated the percentage of AO (concentrations ranging from 200 to 400 nM) internalized by HaCaT cells after the incubation regimes of 10 or 60 min. After the respective incubation period, the cells were lysed with the non-ionic surfactant Triton X-100 and the fluorescence of incorporated and non-incorporated AO recorded. The final percentage of AO internalized was calculated through Equation (1), and the values are reported in Table 2.

**Table 2.** Incorporation of AO (%) in HaCaT cells after AO treatment (200–400 nM) and followed by incubation for 10 or 60 min.

| [AO], nM | Percentage of AO Incorporation in HaCaT Cells | |
|:---:|:---:|:---:|
|  | 10 min | 60 min |
| 200 | 58 ± 4 | 64 ± 8 |
| 300 | 61 ± 6 | 70 ± 5 |
| 400 | 60 ± 7 | 74 ± 3 |

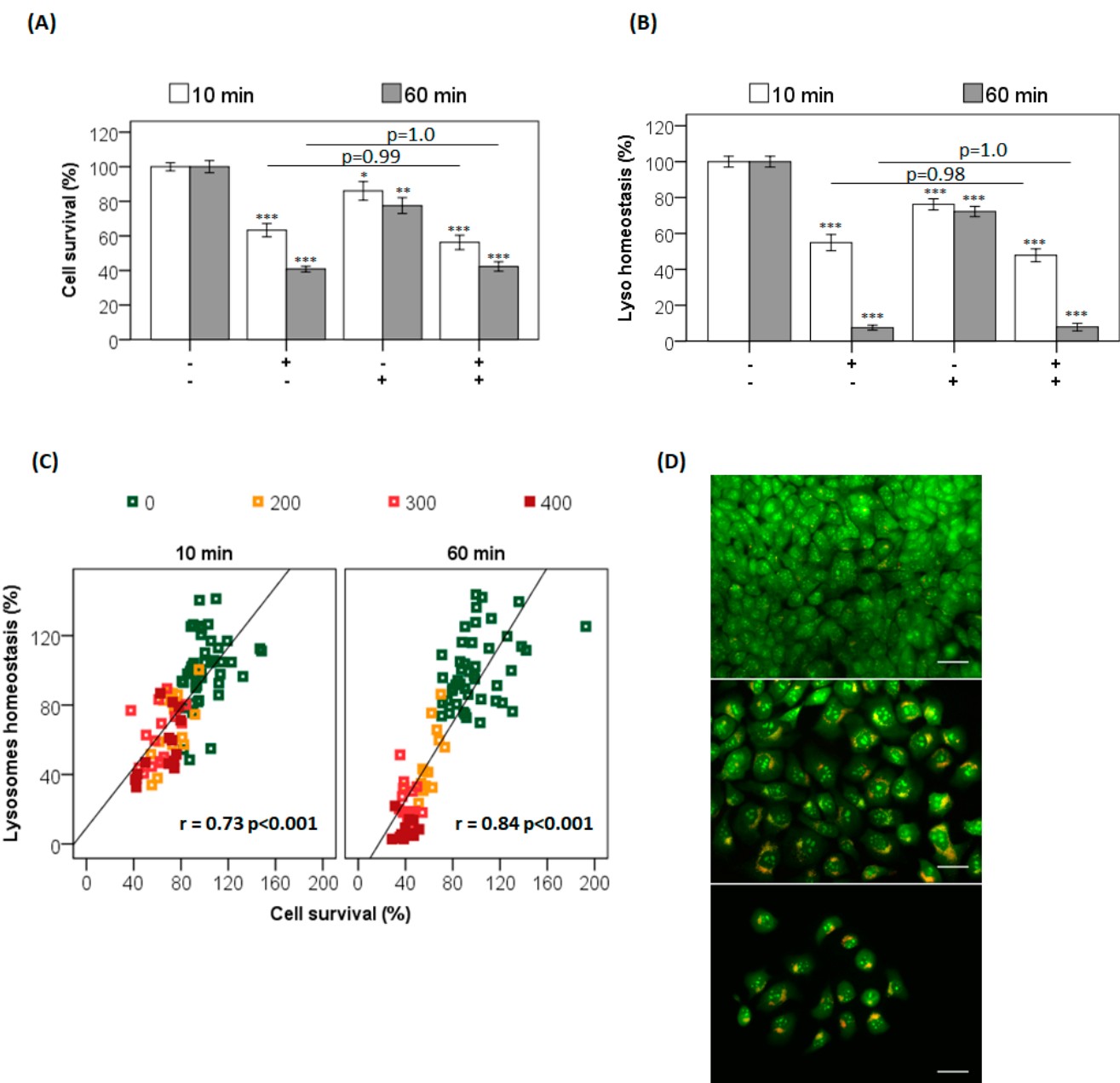

**Figure 3.** (**A**) Crystal violet and (**B**) NRU cell survival according to 400 nM AO treatment, in the absence (−) or presence (+) of additional treatment with the lysosomal inhibitor chloroquine (CQ, 20 μM for 24 h). (**C**) Correlation analysis of lysosomal homeostasis and cell survival according to incubation with AO for 10 or 60 min followed by blue light photosensitization. (**D**) Epifluorescence microscopy of AO-labeled cells for the visualization of acid vacuoles 48 h after photodamage with 400 nM AO for 10 min or 60 min incubation and the corresponding control. Scale bar is 50 μm. ANOVA analyzes: * $p < 0.05$, ** $p < 0.01$, *** $p < 0.001$.

Note that the percentage of AO incorporation into the intracellular environment is between ~60% and 70%, irrespective of the AO concentration and incubation time (Table 2). After a 10 min incubation, the observed AO incorporations were 58, 61 and 60% for 200, 300 and 400 nM of AO, respectively. After a 60 min incubation with AO at 200 nM, the percentage of AO incorporation remained in the same range (60%), showing no statistically significant difference when compared with the incorporation after a 10 min incubation. When the AO concentration was raised to 300 and 400 nM, there was a small increase in the percentage of dye incorporation (around 10%, with statistical significance for 300 nM,

$p < 0.01$, and 400 nM, $p < 0.001$). However, these changes are very small compared to the robust differences in the photocytotoxicity of AO that were observed comparing the shorter and longer incubation times. Although the incubation period did not cause significant differences in terms of total dye incorporation, it might substantially affect the intracellular location of AO. Indeed, we and others have proposed that the intracellular location of the photosensitizer can have significant impact in the photocytotoxicity of any given photosensitizer [3,4,46].

To better understand the targeting of the intracellular AO photodamage according to the incubation time, HaCaT cells were evaluated by epifluorescence microscopy for the incorporation pattern (Figure 4). Note that the cells did not show characteristic orange/red fluorescence staining, because this type of color contrast becomes evident only at much higher AO concentrations [15,42]. The first and most evident difference was that cells treated with AO for the longer incubation time had a higher emission intensity, i.e., were a lot brighter (~3 fold, based on the average integrated areas calculated for cells in Figure 4) than cells incubated with AO for 10 min. Using Image J Software to compare the profiles of fluorescence intensities (Figure S8), it is evident that the AO subcellular localization was more punctual after a 10 min incubation. After a 60 min incubation, AO accumulated in more than one intracellular locus, while a 10 min incorporation seems to target preferentially the EL system (Figure S8, tracing profiles).

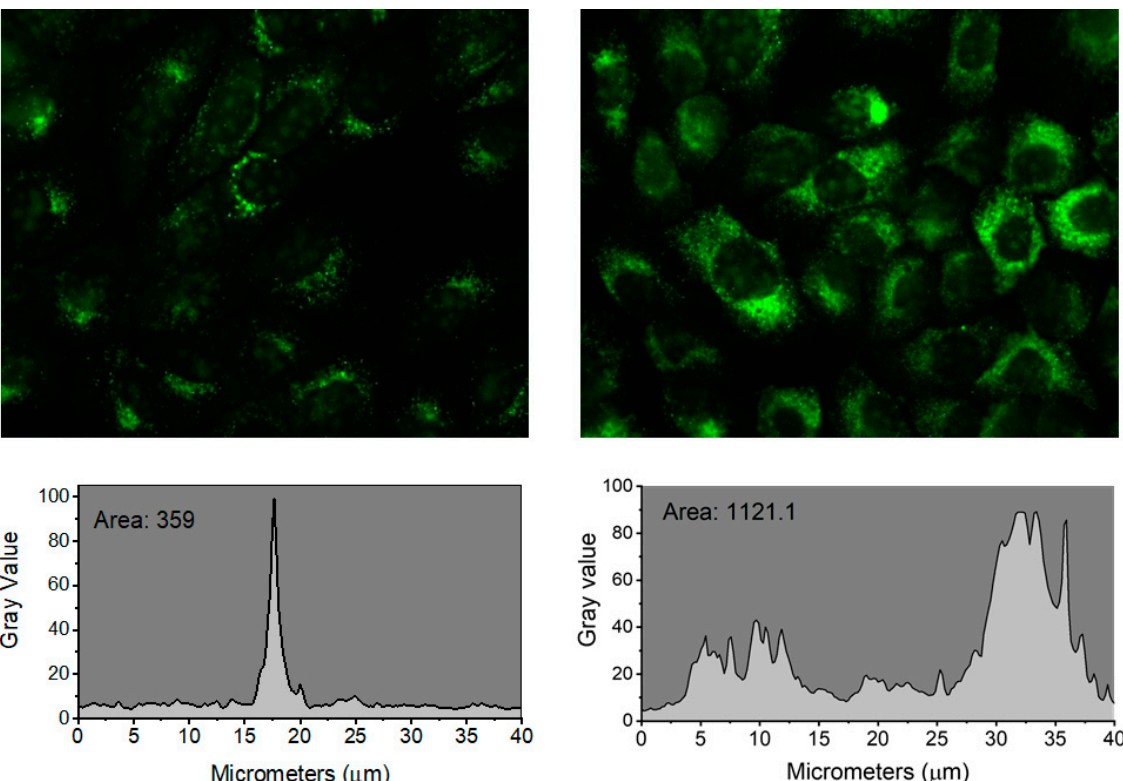

**Figure 4.** Top panels: Epifluorescence microscopy of HaCaT cells labelled with 400 nM AO for visualizing incorporation after 10 min (**left**) or 60 min (**right**) of incubation. Bottom panels: Fluorescence profiles of a single cell in each condition, showing the distinct distribution and intensities. The gray value axis represents the relative fluorescence intensity.

Time-resolved fluorescence lifetime imaging microscopy (FLIM) gives additional evidence of the different environments experienced by AO when the 10 and 60 min incubation times are compared (Figure 5). The bimodal distribution for the longer incubation time, compared with the monomodal distribution for the shorter incubation time, are also evident in the tracing profile of the FLIM image (Figure S9). In addition, the incubation time

clearly affects the AO lifetime. At 10 min a longer fluorescence lifetime predominates (staining in green, ~5 ns), while after 60 min of incubation, the AO accumulation points to a somewhat shorter lifetime (more towards the blue, ~3 ns). Therefore, the intracellular distribution pattern demonstrates significant alterations of the AO subcellular localization and microenvironment depending on the incubation time.

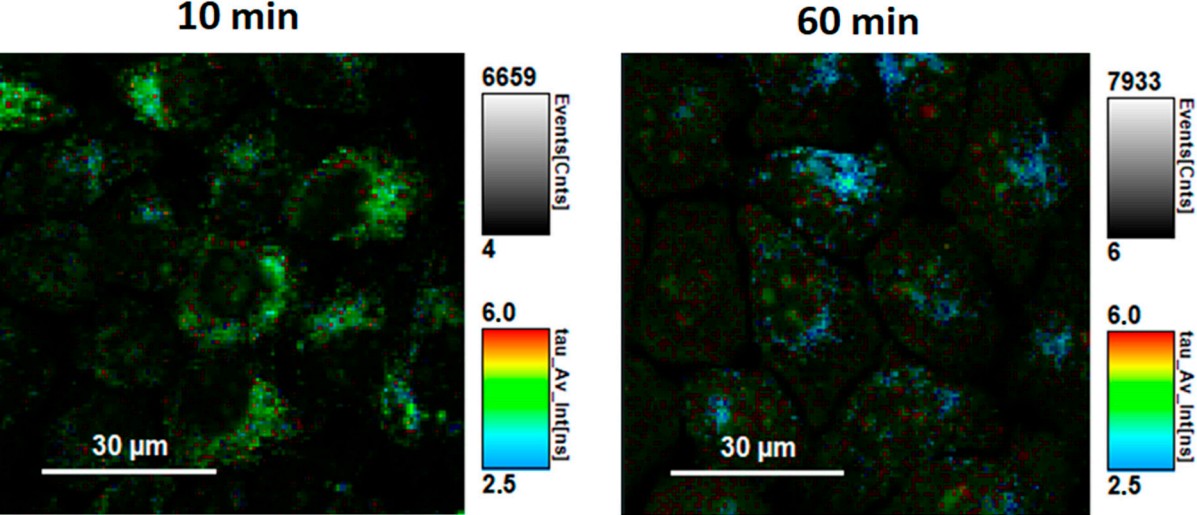

**Figure 5.** Fluorescence lifetime imaging microscopy (FLIM) of HaCaT cells treated with 400 nM AO for 10 (**left**) or 60 (**right**) minutes. The fluorescent species present at the longer incubation time exhibit slightly faster decay than those present after the shorter incubation period.

The efficiency of PDT relies to a large extent on the ability of the photosensitizer to produce an adequate amount of reactive species at the appropriate intracellular location [2,3]. This production can, however, be dependent on the equilibria between monomers, dimers, trimers, and higher aggregates, since aggregation can affect the PS photophysical properties, as shown for many other molecules and systems [47–49]. For the analyses of the monomer-dimer equilibrium at different pHs (acidic or neutral) we have designed an assay using the surfactant SDS. Although we cannot really calculate the relative amounts of dimers and monomers in the intracellular domain, we can evaluate the behavior of AO at the interface of a simple anionic surfactant [50]. To illustrate the tendency of AO to form monomers or dimers at different pHs in the presence of SDS, AO solutions were prepared in different buffers (phosphate buffer and acetate buffer, with equal molarities) and titrated with SDS in the micromolar and millimolar concentration regimens (Figure S10). Because the maximum absorption of monomers and dimers of AO are at 490 nm at 470 nm, respectively, the ratio of the absorbances, $A_{490}/A_{470}$ (M/D ratio), known to affect its photophysical properties [33,51], was plotted against the concentration of surfactant. Figure 6 shows the resultant effects of the addition of the negatively charged SDS and the pH of the aqueous solution on the ground state equilibria (binding to SDS and dye aggregation) experienced by AO. The concentration of monomers and dimers were quantified at each studied condition and expressed as the M/D ratio. At neutral pH, the ratio of M/D decreased rapidly with the increase in the SDS concentration from 0 and 1 mmol L$^{-1}$. This shows that dimers are favorably formed under neutral conditions. In a more acidic environment, a greater fraction of AO is protonated, AOH$^+$, increasing the electrostatic repulsions between AO molecules and disrupting dimer formation. This elucidates the nature of AO monomers and dimers close to surfaces and at different pHs, in particular in conditions such as those experienced by AO in the lysosomes of cells. This is particularly important since AO dimers are less photoactive and have lower phototoxicity.

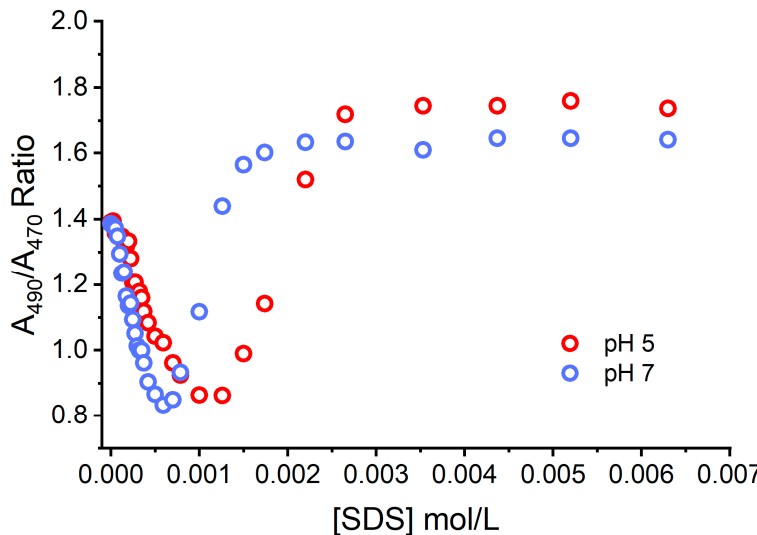

**Figure 6.** Ratio of the absorbances recorded at the maxima of the monomer band, 490 nm, and the dimer band, 470 nm, during titration with 50 mmol L$^{-1}$ SDS in different buffer solutions (pH 7, blue symbols; and pH 5, red symbols).

The correlation between the subcellular localization of a PDT photosensitizer and the final photodynamic efficiency has been the subject of several previous studies. Kessel and co-workers (2003) reported studies of two porphyrins derivatives, DADP-a and DADP-o, which are mitochondrial and lysosomal targeted, respectively; DADP-a required a 2-fold increase in total light dose to achieve the LD$_{90}$ for murine L1210 cells [52]. By comparing the efficiency of two photosensitizers that differ in their singlet oxygen quantum yield by two orders of magnitude, Oliveira and co-workers (2011) quantitatively proved the major role that mitochondrial localization of the PS has on the efficiency of photoinduced cell death [4]. Tsubone and co-workers (2017) confirmed that lysosomes are a better site for photodamage in terms of decreasing the proliferation of tumor cells, compared with mitochondria [24]. Martins and collaborators (2019) showed that parallel damage in lysosomes and mitochondria increased the efficiency of a PDT protocol by two orders of magnitude [41]. Hence, targeting specific subcellular compartments can increase PDT efficiency by generating ROS at specific sites [53–56].

The results obtained in the SDS experiments indicate that the local PS concentration also affects the properties of the PS, since the charge and pH of the organelle will also affect important ground state equilibria such as the monomer/dimer equilibrium. At close to neutral or at acidic pHs, AO is cationic and interacts efficiently with the anionic surfactant SDS. At low surfactant concentrations, AO induces the premicellar aggregation of SDS, at higher SDS concentrations monomers are favored due to the formation of surfactant-stabilized AO-SDS ion pairs. The resultant effects on the absorbance, fluorescence and singlet oxygen generation by AO are shown in Figure 7. The most pronounced contrast appears at the smaller SDS concentrations (Figure 7A–C), showing a higher peak of the monomer band in acid conditions (also observed in the titration in Figure 6). The higher fluorescence emission (Figure 7B) and the increase the singlet oxygen formation (Figure 7C) are attributed to the predominance of monomers at this pH. At an SDS concentration where micelles are present (50 mM SDS), only monomers are present at both pHs (Figure 7D) and the spectral differences disappear (Figure 7E,F). These results provide insight into the possible differences in singlet oxygen generation between monomers and dimers at different proton concentrations, suggesting that photoactive monomers are more prevalent in the acidic environment, such as those found in the lumen of the EL system. However, because lysosomes are a primary target for AO, dimer and aggregate formation inside the lysosomes due to a higher local concentration could potentially hamper their photoactivity.

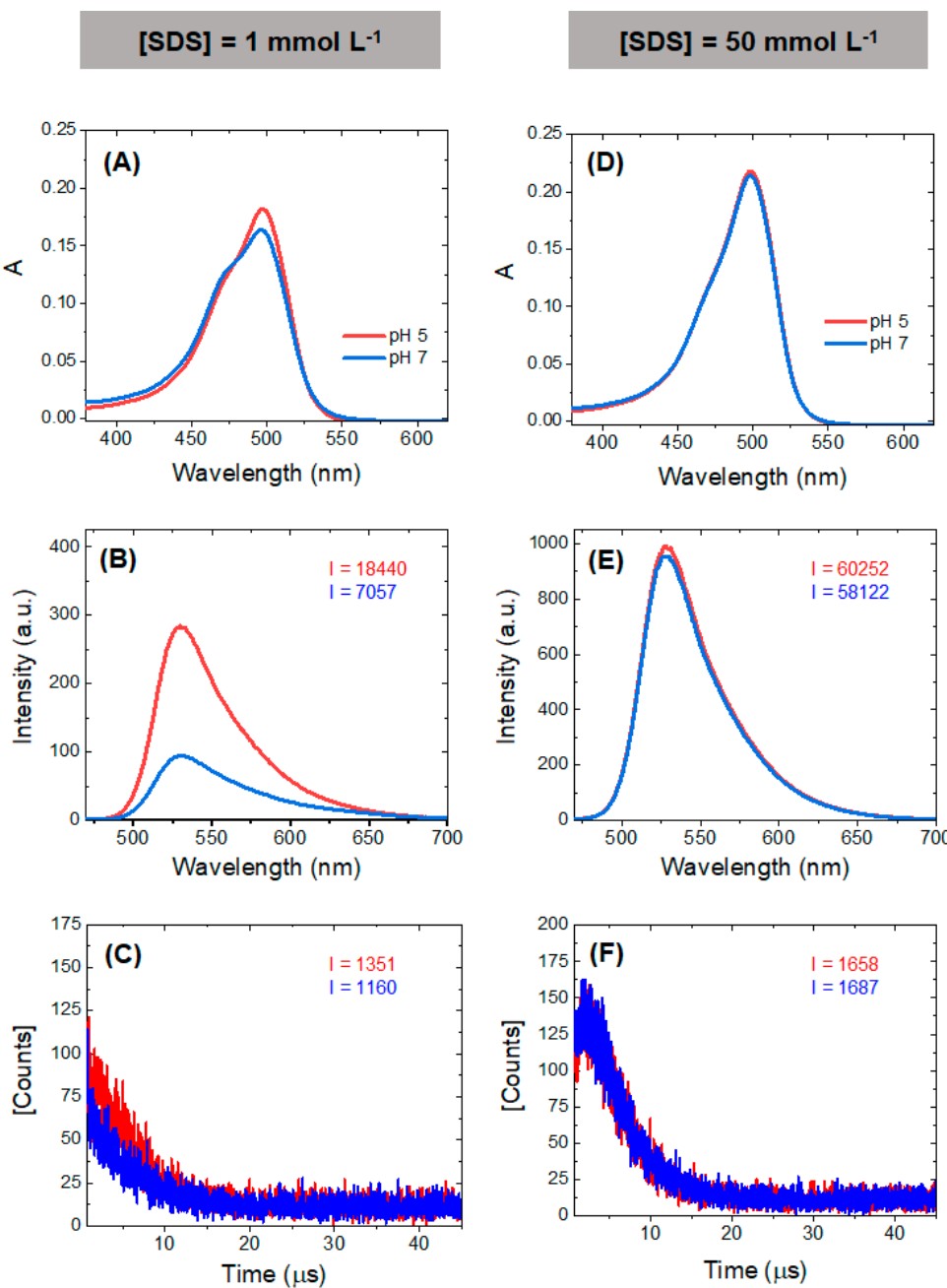

**Figure 7.** Absorbance spectra (**A,D**), fluorescence spectra (**B,E**) and $^1O_2$ luminescence decay (**C,F**) from AO solutions at two pHs (7.4 and 5.0), [AO] = 5 µmol $L^{-1}$ in all solutions. (**A–C**) were performed in [SDS] = 1 mmol $L^{-1}$ and (**D–F**) in [SDS] = 50 mmol $L^{-1}$. The area of each of the integrated signal is represented by "I" inside the boxes. Values of the integrated area under the signal were calculated with Origin 2020 Software. Blue (pH 7) and red (pH 5).

Taken together, the results suggest that the longer incubation time affects the intracellular location and favors the presence of AO as the monomeric species, with higher fluorescence intensity and singlet oxygen generation. There are studies that indicate that singlet oxygen generation in smaller amounts could trigger programmed cell death pathways, while higher $^1O_2$ production may induce cell death by necrosis [57], which is consistent with our results for cells incubated for shorter and longer periods (Figures S5 and S6).

The results from the studies in the presence of SDS are compatible with viability assays, showing that the redistribution of AO favors the monomeric species. Thus, the 60 min incubation greatly negatively impacted cell homeostasis since monomers are more

photochemically active. The yield of $^1O_2$ cannot be solely responsible for the phototoxicity since the lack of a specific intracellular target results in less photodamage [41]. Since the main oxidative species do not live long enough to travel over long distances inside the cell, the amount of ROS produced by the PS does not have as much of an impact on the results of a PDT treatment as does the specific site where these species are being generated. Furthermore, aspects such as symmetry restrictions, charge distribution and axial ligands can alter the targeted organelle, and consequently exert effects on cell viability [52,58]. Hence, targeting one or two specific subcellular sites can increase the PDT efficiency by generating ROS in strategic cell compartments.

When aiming for improved cancer theragnostics, different approaches can be used or combined. These routes seek to achieve efficient results without diminishing the patient's well-being and include tailoring PSs with appropriate substituents [59], drug delivery strategies [60,61], protein-binding [62], photocatalysis [63], and more. Our work provides increased understanding of key points of the application of AO as a PS in PDT-protocols, by unravelling photophysical and photobiological properties that were previously unexplored in depth and that bear on important properties of photosensitizers.

## 4. Conclusions

Besides acting as a widely used cell staining agent, AO has the classical features of an efficient PDT photosensitizer, killing cells in a concentration and light-dose dependent manner. Although the capacity of AO to act as a photosensitizer was known, there was still an almost complete lack of information regarding the mechanisms involved. In the present work, a striking increase in the photodynamic efficiency of AO was observed with increasing incubation time, reported here for the first time, and a reasonable mechanistic explanation is given for this phenomenon. The substantial gain in the photodynamic efficiency of AO at longer incubation times is attributed mainly to the prevalence of monomeric species under this condition, favoring the generation of singlet oxygen by Type II photosensitization. Lysosome damage and autophagy blockage were correlated with cell death at both incubation times, but the severity of the damage was enhanced at the longer incubation time (60 min). The differences in AO uptake by HaCaT cells observed between the two incubation periods were much too small to explain the differences observed. Moreover, steady-state and time-resolved fluorescence imaging proved that AO experiences different intracellular microenvironments, depending on the period of incubation. By evaluating conditions under which AO produces both higher fluorescence emission and greater singlet oxygen generation, it was shown that the longer incubation time favored: 1. dye redistribution to other intracellular domains, and 2. the predominance of the monomeric species, which has a higher photodynamic efficiency. Since AO displays a Type II mechanism of action, from the standpoint of singlet oxygen generations, this redistribution favors the formation of the active species, since lysosomes might in fact initially favor AO dimers. In the context of cancer therapy, AO has an enormous advantage as a PS since the pH of malignant tissues is more acidic, which can contribute to a preference for the formation of monomeric $AOH^+$, but further studies would be necessary to confirm the role of aggregation in cancer cells. The consequences of AO photosensitization in terms of blocking autophagy, together with the enhanced efficiency observed for longer incubation times, can be useful for the development of novel PDT protocols. Finally, the present results point to the necessity of caution when employing AO for living-cell and vital tissue staining [14,64], because exposure of AO to light is likely to induce changes in the cells or tissues by triggering undesirable photosensitized oxidation reactions.

**Supplementary Materials:** The following supporting information can be downloaded at: https://www.mdpi.com/article/10.3390/photochem3020014/s1. Figure S1. Transients of $^1O_2$ emission decays from acridine orange and methylene blue measured in ethanol. Figure S2. Viability test through the reduction of MTT by HaCaT cells for AO-PDT irradiated with blue light ($\lambda = 490$ nm) in the absence and presence of sodium azide. Figure S3. Cell viability assay by reduction of MTT by HaCaT cells, treated with AO for 10 min and 60 min; Figure S4. Lethal doses ($LD_{50}$) of AO

incubated in HaCaT cells for 10 min (left) or 60 min (right); Figure S5. Fluorescence microscopy of HaCaT cells stained with 2.5 μM of AO 48 h after PDT application with 10 min incubation with AO for identification of AVOs; Figure S6. Fluorescence microscopy of HaCaT cells stained with 2.5 μM of AO 48 h after PDT application with 60 min incubation with AO for identification of AVOs; Figure S7. Co-localization of HaCaT cells marked with AO or LDR at 10 min of incubation; Figure S8. Epifluorescence images of HaCaT cells incubated with 400 nM AO for 10 min (top) and 60 min (bottom); Figure S9. I Profiles of fluorescence intensities from HaCaT cells treated with 400 nM AO for 10 min (solid line) and 60 min (dashed line), obtained from FLIM images; Figure S10. Absorption and fluorescence spectra of AO titrated with 50 mmol L$^{-1}$ SDS at different pHs (5 and 7).

**Author Contributions:** Concept of the article, M.S.B. and W.K.M.; experimental analysis, B.F., M.S.J. and H.C.J.; writing—original draft preparation, B.F., W.K.M. and M.S.B.; writing—review and editing, B.F. and M.S.B.; supervision, M.S.B. All authors have read and agreed to the published version of the manuscript.

**Funding:** This research was funded by CEPID REDOXOMA FAPESP # 2013/07937-8.

**Institutional Review Board Statement:** Not applicable.

**Informed Consent Statement:** Not applicable.

**Data Availability Statement:** The data presented in this study are available on request from the corresponding author.

**Acknowledgments:** B.F. and M.S.J. acknowledge CAPES for their Ph.D. and MSc fellowships, respectively. M.S.B. acknowledges CNPq for the productivity fellowship. We thank Frank H. Quina for the English revision, Divinomar Severino for helping B.F. in the interpretation of the experimental results and NAP-PhotoTech—Research Consortium for Photochemical Technology at USP, for maintaining and access to the photophysics/photochemical equipment.

**Conflicts of Interest:** The authors declare no conflict of interest.

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
