# Peer review of "Photodynamic Activity of Acridine Orange in Keratinocytes under Blue Light Irradiation"

_2673-7256, doi:10.3390/photochem3020014_

Round 1

Reviewer 1 Report

Comments to the Author:

In this manuscript, the authors presented different incubation times at the same OA concentration significantly affects the AO phototoxicity. Although the results had demonstrated a substantial gain in the photo- dynamic efficiency of AO at longer incubation times. I think this article is not innovative enough. I reject this article.

Suggested revisions and questions are listed in the following:

1. Page5 line 204 “For example, AO dimer has characteristic maxima of absorption and emission at 470 nm”. Are the maxima characteristic absorption and emission peaks of AO same?

2.The manuscript is poorly written, the language and particularly the grammar require further improvement. For example, page 11 line 362 “Because the maximum absorption of monomers and dimers of AO are 362

respectively at 490 nm at 470 nm” should be “Because the maximum absorption of monomers and dimers of AO are at 490 nm and 470 nm, respectively”.

3. Please add the space between unit and number in full manuscript. For example, page 5 line 216 “298K” should be “298 K”.

Reviewer 2 Report

This manuscript investigated the PDT mechanism of Acridine orange (AO), which has not been reported before. The results are interesting, however, there should be more experiments to support the mechanism of AO in cells.

1. Which pathway— Type-I or Type-II—did AO use to cause the production of ROS? Have you got any proof? 

2. Is the light dose used in this experiment suitable for PDT treatment?

Reviewer 3 Report

In the present work Fornaciari and colleagues present a study of the photodynamic properties of acridine orange. In particular this dye, which is employed in biological cellular assays, is showed to have toxic effects when employed in certain conditions (concentration, incubation time, etc…). The authors nicely provide references on the proper conditions in which such assays should be performed.

The data are in general well described and presented; however, to be suitable for publication in Photochem the authors must absolutely reply to the following points:

1. The authors discuss on the internal localization of acridine orange at 10 and 60 minutes of incubation. I agree with them that the localization in cell cytosolic subcompartments varies with time, being much more distributed in the various intracellular compartments at 60 min, but the localization and discussion is extremely speculative.

Can the authors provide any colocalization of acridine orange with dyes that localize in specific cellular subcompartments, for example lysososmes, mitochondria or endoplasmatic reticulum?

2.       When discussing photodynamics, the authors should report recent relevant literature which is missing at present in the references, as for example: 

i) Huaiyi Huang, et al. Nature Chemistry 2019, 11, 1041 (https://doi.org/10.1038/s41557-019-0328-4)

ii) Huayun Shi, et al. British Journal of Cancer 2020, 123, 871 (https://doi.org/10.1038/s41416-020-0926-3)

iii) Cantelli Andrea, et al. JACS Au 2021, 1, 925 (https://doi.org/10.1021/jacsau.1c00061)

Reviewer 4 Report

please see the attached file.

Round 2

Reviewer 1 Report

I accept your suggestions for revision. Through your revision of the article,we have clearer understanding of the novelties of your work.

Author Response

Dear Reviewer, we thank you for your careful revision, we have sent the manuscript for a full-length English revision, and we hope the language is appropriate in this new manuscript version.

Reviewer 2 Report

Accept.

Author Response

We thank the reviewer for the careful revision of the manuscript.

Reviewer 3 Report

The authors clarified the points raised in the previous review, and added new experiments. However the new data presented (Figure S7) for DAPI (panel C) are not clear: why the signal from DAPI (i.e. DNA) is not located only in the nuclei? It seems overlapped with the signal of panel A and not coming from nuclei and DNA. The authors must clarify this point.

Author Response

We thank the reviewer for the comments and revision. We did not fully understand the question by the reviewer, since Figure S7 presents the fluorescence intensities of HaCaT cells incubated with acridine orange (AO) and the lysosomal marker Lysotracker Deep Red. Figure S7-C shows the merged co-localization of both mentioned dyes. There is no DAPI staining in the mentioned figure.

Reviewer 4 Report

Line 160

Please Correct from "Photophrin" to "Photofrin"

Author Response

Dear reviewer, we thank you for the suggestion. The appropriate correction was made in the present version of the manuscript.

Round 3

Reviewer 3 Report

Accept in the present form